# Incidence and Risk Factors of Pre-Eclampsia in the Paropakar Maternity and Women’s Hospital, Nepal: A Retrospective Study

**DOI:** 10.3390/ijerph16193571

**Published:** 2019-09-24

**Authors:** Seema Das, Rupesh Das, Rashmita Bajracharya, Gehanath Baral, Bina Jabegu, Jon Øyvind Odland, Maria Lisa Odland

**Affiliations:** 1Department of Community Medicine, The Arctic University of Norway, 9019 Tromsø, Norway; jon.o.odland@ntnu.no; 2Department of Medicine, Janaki Medical College Teaching Hospital, Janakpur 45700, Nepal; drrupeshdash@gmail.com (R.D.); binajabegu@gmail.com (B.J.); 3Department of Sociology and Gerontology, Miami University, Oxford, OH 45056, USA; rashmitabajracharya3@gmail.com; 4Department of Obstetrics and Gynecology, National Academy of Medical Sciences, Kathmandu 44600, Nepal; gehanathbaral@gmail.com; 5Department of Public Health and Nursing, Norwegian University of Science and technology, 7491 Trondheim, Norway; 6Institute of Applied Health Research, University of Birmingham, Birmingham B15 2TT, UK; maria.l.odland@ntnu.no

**Keywords:** pre-eclampsia, risk factors, retrospective study, Nepal

## Abstract

This study aims to determine the incidence of pre-eclampsia and distribution of risk factors of pre-eclampsia at Paropakar Maternity and Women’s Hospital, Kathmandu, Nepal. A retrospective study included 4820 pregnant women from 17 September to 18 December 2017. Data were obtained from the medical records of the hospital’s Statistics Department. Associations between the risk factors and pre-eclampsia were determined using logistic regression analysis and expressed as odds ratios. The incidence rate of pre-eclampsia in the study population was 1.8%. Higher incidence of pre-eclampsia was observed for women older than 35 years (Adjusted Odds Ratio, AOR)= 3.27; (Confidence Interval, CI 1.42–7.52) in comparison to mothers aged 20–24 years, primiparous women (AOR = 2.12; CI 1.25–3.60), women with gestational age less than 37 weeks (AOR = 3.68; CI 2.23–6.09), twins pregnancy (AOR = 8.49; CI 2.92–24.72), chronic hypertension (AOR = 13.64; CI 4.45–41.81), urinary tract infection (AOR = 6.89; CI 1.28–36.95) and gestational diabetes (AOR = 11.79; CI 3.20–43.41). Iron and calcium supplementation appear to be protective. Age of the mothers, primiparity, early gestational age, twin pregnancy, chronic hypertension, urinary tract infection and gestational diabetes were the significant risk factors for pre-eclampsia. Iron and calcium supplementation and young aged women were somewhat protective.

## 1. Introduction

Pre-eclampsia is an increase in systolic and diastolic blood pressure to ≥140 mm of Hg and ≥90 mm of Hg respectively in two or more consecutive occasions (≥4 h apart) after 20 weeks of gestation; the rise in blood pressure is combined with one or more of the following conditions: (1) proteinuria (a 24-h urine collection with a total protein excretion of ≥ 300mg or ≥ 1 + on urine dipstick); (2) evidence of other maternal organ dysfunction such as renal insufficiency (creatinine < 90 umol/L), liver involvement (elevated transaminases or epigastric pain), neurological complications, hematological complications; (3) fetal growth restriction [1]. There are several maternal and clinical risk factors that either alone or in combination may contribute to the high risk of pre-eclampsia; genetic factors, diet, parity, gestational weight gain, maternal age, twin pregnancy, previous history of pre-eclampsia, maternal pre-existing conditions (such as diabetes, chronic hypertension, and infections) are considered to play influential roles in the development of pre-eclampsia [2,3]. Pre-eclampsia is an important cause of maternal, perinatal and neonatal morbidity and mortality; it complicates about 2–8% of pregnancies [4].

Incidence of pre-eclampsia in developing countries is approximately seven times higher than in developed countries (on average 2.8% of live births versus 0.4%) [5,6]. Nepal is a developing country and pre-eclampsia/eclampsia is second leading cause of maternal morbidity and mortality in Nepal [7]. In 2014, Bilano et al. showed that out of 8265 deliveries in Nepal, 180 (2.18%) of women developed pre-eclampsia with adverse maternal and perinatal outcomes; pre-eclampsia is attributable to more than 1% of maternal deaths, approximately 11% perinatal death, and about 25% preterm births and 38% low birthweight in Nepal [8].

To date, no studies have addressed the risk factors for pre-eclampsia in Nepal. There are few studies on risk factors for pre-eclampsia that have been conducted in other developing countries. However, there may exist some inconsistencies between Nepal and other developing countries [8]. Therefore, the present study aims to: (i) determine the incidence of pre-eclampsia in the Paropakar Maternity and Women’s Hospital in Nepal; and, (ii) to examine the associations between pre-eclampsia and the factors such as maternal age, parity, gestational age, twins pregnancy, dietary supplementations during pregnancy and, maternal health issues including gestational diabetes, chronic hypertension, urinary tract infection and asthma.

## 2. Materials and Methods

### 2.1. Sampling

The retrospective study was conducted at the Paropakar Maternity and Women’s Hospital, which is the largest public maternity hospital in Nepal. We retrospectively reviewed the medical charts of all pregnant women who gave birth in the hospital from 17 September to 18 December 2017. Based on an expected prevalence of pre-eclampsia of 2.18% in Nepal [8], a sample size of 4820; a power of 80%; a significance level (*p*-value) of 0.05 and an exposed/non-exposed group ratio of 1:4 [9], odds ratios of ≥ 0.90 were detectable. Due to the low prevalence of diseases and time constraints the ratio 1:4 was selected.

### 2.2. Data Collection

We retrieved information about the patients and newborns from the hospital’s official patient charts. As computerized record was not available, all available information about the patient and their newborn were entered into the study database using Microsoft Excel (Microsoft, NY, USA). A double data entry system was used to minimize errors.

A two-step process was implemented to identify patients with pre-eclampsia. Patients with a recorded diagnosis of pre-eclampsia were first identified. Then, we reviewed patient’s charts for specific clinical and laboratory findings and compared to the WHO criteria (at least 140/90 mm of Hg or above on two occasions at least 4 h apart after 20 weeks of gestation combined with proteinuria > 0.3 g/24 h or ≥1 measured by a urine dipstick) [4]. Information obtained from the medical records about diseases other than pre-eclampsia were included: gestational diabetes mellitus (defined as the increased blood sugar level after 20 weeks of the gestational age, and fasting glucose level ≥ 6.7 mmol/L; urinary tract infection (recorded diagnosis and White blood cells in urine sample and/or urine culture report); hypothyroidism (recorded diagnosis and/or abnormal thyroid function test report); and asthma and sub-fertility treatment (recorded diagnosis).

The information obtained was cleaned, sorted and coded to facilitate data analysis. Maternal age at the time of delivery was categorized as 15–19, 20–24, 25–29, 30–34, and ≥35 years. Parity was defined as the number of previous live births and stillbirths and was dichotomized into primiparity and multiparity. Gestational ages recorded at the time of delivery were classified into <37, 37–41 and >41 weeks. Supplementation included intake of iron and calcium during pregnancy. Pregnancies were designated as singletons and twin pregnancy. Occurrence of pre-eclampsia, chronic hypertension, gestational diabetes, hypothyroidism, sub-fertility treatment, and asthma were recoded as a binary variable with a yes/no response.

### 2.3. Data Analysis

Descriptive analysis was done to evaluate the distributions of the variables in the study and the statistical findings are reported as numbers and percentages. Occurrence of pre-eclampsia is the dependent variable. Initially, bivariable logistic regression analyses were conducted to estimate the unadjusted risk of independent variables on pre-eclampsia. Variables (including potential confounders) with *p*-values <0.05 in this preliminary step were included in the subsequent multivariable logistic regression analysis (employing the enter method). Odds ratios and their 95% confidence intervals were calculated in the multivariable logistic regression to estimate the independent effect of pre-eclampsia risk factors on pre-eclampsia occurrence. The Hosmer-Lemeshow model goodness-of-fit test was applied. Multi-collinearity was tested using the variance inflation factor (VIF) among the independent variables prior to their entry into the model. The data were analyzed using the Statistical Package for Social Science (SPSS) version 24 (IBM, NY, USA).

### 2.4. Ethical Considerations

Ethical approval was obtained from the Nepal Health Research Council (NHRC) and The Regional Committee for Medical and Health Research Ethics Northern Norway (REK Nord-2017/2440). In addition to this, a permission letter was obtained from the director and the research committee of the Paropakar Maternity and Women’s Hospital. All registered data were anonymized prior to their use for statistical purposes and related assessments.

## 3. Results

### 3.1. Incidence Rate and Distribution of Factors Associated with Pre-Eclampsia

A total of 4820 pregnant women of age 15–38 were included in the study, 1.8% (*n* = 85) had a diagnosis of pre-eclampsia with mean (standard deviation) of maternal age and gestational age were 26.4 (5.2) and 38 (16.7) respectively; and 98.2% (*n* = 4735) didn’t have a diagnosis of preeclampsia with mean (standard deviation) of maternal age and gestational age were 24.4 (4.6) and 39 (12.0).The incidence rate of pre-eclampsia in the study population was 1.8% (Table 1). Among patients with pre-eclampsia almost 12% were older than 35 years, 64.7% were primiparous, and 32.9% had gestational age less than 37 weeks (Table 1). Similarly, the proportion of diseases such as gestational diabetes (4.7% versus 0.3%), chronic hypertension (5.9% versus 0.3%) and urinary tract infection (2.4% versus 0.3%) was significantly higher (*p* ≤ 0.01) among women with pre-eclampsia compared to those without (Table 1).

### 3.2. Risk Factors of Pre-Eclampsia

Independent variables which had a statistically significant association with pre-eclampsia in the bivariable analysis, summarized in Table 1, were included in the multivariable analysis. Five factors were predictive of pre-eclampsia in bivariable analysis: age, parity, gestational age, supplementation, twin pregnancy, and maternal diseases. All of these factors were still statistically significant in the multivariable analysis (Table 2); however adjusted odds ratio (AOR) of gestational age, use of iron and calcium supplementations, twin pregnancy, chronic hypertension, urinary tract infection and gestational diabetes were somewhat lower compared to the crude odds ratios. Increased values of adjusted odds ratio were evident for the following: primiparity (by 48.3%), ages 25–29 (34.3%) and 30–34 (11.9 %), and gestational age above 41 weeks (by 22.0%). But these change in multivariable analysis it did not show any significance. Primiparity remained statistically significant with two-fold higher odds (AOR = 2.12; 95% CI, 1.25–3.60) (*p* ≤ 0.01) of developing pre-eclampsia compared to multiparous women in multivariable analysis (Table 2).

Similarly, women aged 35 years and above retained 3.2-fold increased odds of pre-eclampsia compared to 20–24 group. The positive trend in the prevalence of pre-eclampsia with increasing age also remained after adjustment, as did the apparent protective influence of calcium and iron intake during pregnancy. Gestational age below 37 weeks (AOR = 3.68; 95% CI 2.23–6.09), twin pregnancy (AOR = 8.49; 95% CI 2.92–24.72) and the maternal diseases condition variables such as chronic hypertension (AOR = 13.64; 95% CI 4.45–41.81), urinary tract infection (AOR = 6.89; 95% CI 1.28–36.95) and gestational diabetes (AOR = 11.79; 95% CI 3.20–43.41) were again robustly and significantly (*p* = < 0.05) associated with an increased risk for developing pre-eclampsia.

## 4. Discussion

### 4.1. Main Findings

A total of 4820 deliveries were included in this study, and of these, 1.8% developed pre-eclampsia. Both bivariable and multivariable logistic regression analyses show that maternal age above 35 years, gestational age below 37 weeks, twin pregnancy, chronic hypertension, urinary tract infection, and gestational diabetes were risk factors of pre-eclampsia. The multivariable analysis also indicates that the primiparity is significantly associated with an increased risk of pre-eclampsia and age 15–19 years has the protective effect.

### 4.2. Data Interpretation and Comparisons with Previous Findings

The incidence of pre-eclampsia in the present study (1.8%) is comparable with findings from Koshi Zonal Hospital Nepal (1.5%) [10] and WHO Global Survey on Maternal and Perinatal Health (2.18%) [8]. However, the incidence rates of pre-eclampsia in Nepal reported in this study are higher than that reported by the Abalos et al. [11]. The latter authors suggest that lower incidence rates likely occur when strict criteria for defining pre-eclampsia are adhered to and when under-reporting of pre-eclampsia cases occurs [11]. On the other hand, the incidence of pre-eclampsia in the current study is lower than that of the neighboring countries like India (4.0%) and China (2.8%), and higher than that reported for Sri Lanka (1.4%) [8]. The variation of incidence of pre-eclampsia between Nepal and other developing countries might be because of the different distribution of maternal risk factors, availability/accessibility of health services and diagnostic capacities [8,12]. In addition, in low income countries (GNI per capita $995 or less-2017 estimate), the quality of data might be impacted by a lack of funds and therefore of manpower for routine and systematic registration of data.

The risk factors for pre-eclampsia ascertained in this study are similar to the risk factors that have been described in other studies conducted in different countries [3,8,13]. Most of the studies showed that the risk of pre-eclampsia increases with age [8,11,12,14]. Supporting this, our study indicates that pregnant women who were 35 years old had more than 3 times higher odds of developing pre-eclampsia compared to 20–25-year-old women. This could be because of aging of uterine blood vessels and increased arterial stiffness leads to gradual loss of compliance of the cardiovascular vessels causing endothelial dysfunction (a characteristics of preeclampsia) [14,15]. As reported by Al-Tairi et al. [16] our study also found that women aged between 15–19-years-old experienced protection against pre-eclampsia relative to the 20–24 age group which is in contrast to the studies conducted in Indonesia [17] and India [18]. This might be due to the effects of confounders, type of study design and studied population diversity.

Studies have identified that primiparity is a risk factor for this outcome [8,12,13,17]. In line with this, our study also observed that primiparous mothers had two-fold higher risk for developing pre-eclampsia than multiparous women. These findings are expected as the first exposure to chorionic villi [which is of fetal origin] and related maternal immunological incompetence are more likely during the first pregnancy and can increase risk of pre-eclampsia [15].

Our study demonstrates that gestational age below 37 weeks is significantly associated with pre-eclampsia. However, women with pre-eclampsia have a higher probability of giving pre-term birth compared to women without pre-eclampsia [19]. Hence, preterm delivery is the consequence of preeclampsia, and lower gestational age cannot be indicated as a risk factor of preeclampsia.

Studies conducted in India [8,20,21], Taiwan [13] and 29 low-and-middle-income countries [11] have revealed that women who have given birth to twins, triplets, or multiple fetuses are more likely to develop pre-eclampsia compared to the singleton pregnancies. In agreement, our study indicates that twin pregnancy enhanced the odds eight-fold of developing pre-eclampsia compared to the singleton pregnancies. It has been suggested that the increased risk of pre-eclampsia development during multi-fetal pregnancies might be due to the large placental mass and increased circulating levels of soluble fms-like tyrosine kinase-1(sFlt-1); the latter leads to high soluble fms-like tyrosine kinase-1 to placental growth factor (PlGF) ratios [22] and may be taken as predictor of pre-eclampsia [16].

Underlying medical conditions such as chronic hypertension, gestational diabetes, urinary tract infection is found to be associated with increased risk of pre-eclampsia [8,11,18]. Our study found that chronic hypertension had nearly 14-fold higher odds of pre-eclampsia, which is in line with studies conducted in India [18,21], Yemen [16], Ethiopia [14], Jordan [12], Uganda [23] and the WHO Global Survey on Maternal and Perinatal Health study [8,11]. Elevated cardiac output and increased systematic vascular resistance in hypertension is suspected to lead to endothelial cell dysfunction [15]. Gestational diabetes patients had nearly 12-folds higher odds of pre-eclampsia and this is supported by the other studies [8,24]. This is biologically plausible because insulin resistance and high levels of insulin cause increased sympathetic activity and abnormal tubular sodium absorption, which eventually lead to endothelial cell damage and thus increased risk of pre-eclampsia [15]. Some studies [8,13] reported that the pre-eclampsia is more common in women who have urinary tract infection, and thus our observation of a near 7-fold increase concurs. It has been suggested that this impact may be due to the increased inflammatory response during infectious diseases [25]. Our bivariable analysis suggests that hypothyroidism, asthma and sub-fertility treatment are associated with an increased risk of pre-eclampsia. Asthma and sub-fertility treatment did not meet the statistical significance and were not included in the multivariable analysis. Hypothyroidism showed statistically significant association with pre-eclampsia but not included in multivariable analysis as no research has been reported about hypothyroidism as a predictor of pre-eclampsia. Additional research needs to be carried out to establish hypothyroidism as a predictor of pre-eclampsia.

### 4.3. Strengths and Limitations

This is the first known retrospective study conducted in Nepal at the Paropakar Maternity and Women’s Hospital. As maternal mortality is comparatively high in Nepal and pre-eclampsia is a well-known contributor to maternal death, it is important to identify the risk factors for pre-eclampsia to improve intensive care. The sample size was large and had adequate power to estimate associations in a multivariable analysis.

As this study was limited to only one hospital in Nepal, the findings may not be generalizable to the national population of pregnant women. Although most of the babies born in an institutional setting are delivered at the study hospital, the national average of institutional deliveries is only 55.1%; therefore, the study results will not be applicable to pregnant mothers who never come to hospital for check-ups and give home births [26]. Also, limiting our study to only a single hospital may have caused unintended selection bias. Likewise, as the data is non-digitalized in nature, that could have increased the chance of lost or unavailability of patient’s record chart also leading to selection bias.

Even though the multivariable analysis was carried out, many potential confounders were not controlled, due to a lack of pertinent information; for instance, socio-demographic data such as ethnicity, level of education, occupational status, residence area, body mass index, smoking status, and maternal factors such as maternal obesity, gestational weight gain, type 1 and type 2 diabetes mellitus, and maternal history of previous preeclampsia to list a few. Similarly, our data were extracted from the hospital’s official paper bound record book and patient charts, which increases the possibility of clerical errors that could have led to information bias.

## 5. Conclusions

This study showed that there are various modifiable and non-modifiable risk factors of pre-eclampsia, such as maternal age, primiparity, gestational age, twin pregnancy and maternal diseases including chronic hypertension, urinary tract infection and gestational diabetes. Therefore, the result of this study will be useful for policy and clinical purposes, as understanding the determinants of pre-eclampsia will facilitate the prioritization of interventions and thus resource allocations. In addition, it will identify high-risk pregnancies (those in which the health and life of both mother and fetus are at increased risk) [8]. This helps to achieve substantial improvement in maternal, perinatal, and neonatal health of Nepal.

## Figures and Tables

**Table 1 ijerph-16-03571-t001:** Distributions of factors associated with pre-eclampsia and Bivariable logistic regression analysis outcomes.

Independent Variables	Pre-Eclampsia *n* (%)		Bivariable Analysis		
	Yes 85 (1.8)	No 4735 (98.2)	Crude Odds Ratio	95% Confidence Interval (CI)	*p*-Value
Age (years)					
15–19	4 (4.7)	610 (12.9)	0.36	0.12–1.01	0.053
20–24	38 (44.7)	2084 (44.0)	1	Reference	-
25–29	18 (21.2)	1353 (28.6)	0.73	0.42–1.28	0.27
30–34	15 (17.6)	518 (10.9)	1.59	0.87–2.91	0.13
≥35	10 (11.8)	170 (3.6)	3.23	1.58–6.59	0.001
Parity					
Primiparity	55 (64.7)	2661 (56.2)	1.43	0.91–2.24	0.12
Multiparity	30 (35.3)	2074 (43.8)	1	Reference	-
Gestational age					
<37 weeks	28 (32.9)	466 (9.8)	4.34	2.72–6.92	0.000
37–41 weeks	54 (63.5)	3903 (82.4)	1	Reference	-
>41 weeks	3 (3.5)	366 (7.7)	0.59	0.18–1.90	0.38
Iron and calcium supplements	75 (88.2)	* 4518 (95.4)	0.08	0.04–0.16	0.000
Twin pregnancy	5 (5.9)	29 (0.6)	10.14	3.82–26.87	0.000
Maternal Diseases					
Chronic Hypertension	5 (5.9)	15 (0.3)	19.67	6.97–55.41	0.000
Urinary tract infection	2 (2.4)	15 (0.3)	7.58	1.70–33.69	0.008
Gestational diabetes	4 (4.7)	13 (0.3)	17.93	5.72–56.12	0.000
Hypothyroidism	4 (4.7)	39 (0.8)	5.95	2.08–17.03	0.001
Sub-fertility treatment	1 (1.2)	14 (0.3)	4.01	0.52–30.88	0.182
Asthma	1 (1.2)	9 (0.2)	6.25	0.78–49.89	0.084

* Missing = 169 (3.6%).

**Table 2 ijerph-16-03571-t002:** Multivariable analysis of factors associated with pre-eclampsia.

Age	Independent Variables		
	Adjusted Odds Ratio (AOR)	95% Confidence Interval (CI)	*p*-Value
15–19	0.25	0.08–0.74	0.01
20–24	1	Reference	-
25–29	0.98	0.53–1.77	0.94
30–34	1.78	0.87–3.65	0.12
≥35	3.27	1.42–7.52	0.005
Parity			
Primiparity	2.12	1.25–3.60	0.005
Multiparity	1	Reference	-
Gestational age			
<37 weeks	3.68	2.23–6.09	0.00
37–41 weeks	1	Reference	-
>41 weeks	0.72	0.22–2.32	0.58
Iron and calcium supplements	0.062	0.03–0.14	0.00
Twins pregnancy	8.49	2.92–24.7	0.00
Maternal Diseases			
Chronic hypertension	13.6	4.45–41.8	0.00
Urinary tract infection	6.89	1.28–37.0	0.02
Gestational diabetes	11.8	3.20–43.4	0.00

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
