# Peer review of "Incidence and Risk Factors of Pre-Eclampsia in the Paropakar Maternity and Women’s Hospital, Nepal: A Retrospective Study"

_ijerph, 2019, doi:10.3390/ijerph16193571_

Round 1
Reviewer 1 Report
The manuscript focuses on one of the most important obstetric problems (i.e. preeclampsia, PE) and its risk factors. The study included 4820 pregnant women, who were hospitalized at the largest public maternity hospital in Nepal. The authors calculated the incidence of PE in this group and selected a few variables for determining their predictive values.
There are some limitations:
Abstract:
- line 17: "a total of 4820 deliveries" - there is an inaccuracy. The authors informed about "4820 pregnant women" (line 111) and multiple pregnancies (5 in the PE group and 29 in the non-PE group; Table 1). Please correct the number of deliveries in the line 17 or change the sentence, for example: "A retrospective study included 4820 pregnant women" instead of : “was conducted with a total of 4820 deliveries”.
Materials and Methods
- line 87: The authors informed that multiple pregnancies were grouped as twins, triplets or multi-fetal pregnancies". The results did not include these data.
- The authors mentioned iron and calcium supplementation during pregnancy. What about low-dose aspirin prophylaxis? How many women received this therapy for prevention of PE, especially in the case of pregnant women with at least one high-risk condition (for example: previous pregnancy complicated by PE, multiple gestation, chronic hypertension, type 1 or 2 diabetes mellitus, chronic kidney disease, autoimmune disease)?
- The main limitation of this study is the fact that the authors selected few variables as potential risk factors, i.e. maternal age, parity, gestational age, supplementation (of iron and calcium), multiple pregnancy, and six maternal conditions (chronic hypertension, urinary tract infection, gestational diabetes mellitus, hypothyroidism, sub-fertility treatment, asthma). Is it possible to calculate additional parameters, for example maternal obesity, gestational weight gain, type 1 and 2 diabetes mellitus, maternal history of previous PE, etc.
Results and Discussion:
- Table 1 - Is it true that p-value means 0.000 for the age 20-24 (if there are: 44.7% vs. 44%). Similarly, Table 2 - p-value 0.002 for the age 20-24 (?)
- line 130 - Age 15-19 years is a protective factor in the study, so this variable could not be indicated as a risk factor of PE.
- line 130 as well as lines 183-185 - lower gestational age in the PE group could not be indicated and discussed as a risk factor of PE. This is a consequence of PE. If the diagnosis is made, the definitive treatment is delivery to prevent development of maternal or fetal complications from disease progression. Delivery results in eventual resolution of the disease. Timing of delivery is based upon a combination of factors, including disease severity, maternal and fetal condition, and gestational age. Could you specify the time of PE diagnosis? Could you include the percentage of severe PE in the PE group?
Author Response
Dear reviewer,
Thank you for your thoughtful feedback. We have responded to your all the comments below and have made significant changes to the manuscript.
Response to Reviewer 1 Comments
Point 1: Abstract:
- line 17: "a total of 4820 deliveries" - there is an inaccuracy. The authors informed about "4820 pregnant women" (line 111) and multiple pregnancies (5 in the PE group and 29 in the non-PE group; Table 1). Please correct the number of deliveries in the line 17 or change the sentence, for example: "A retrospective study included 4820 pregnant women" instead of: “was conducted with a total of 4820 deliveries”.
Response 1:
We have accepted your suggestion to change “was conducted with a total of 4820 deliveries” to "A retrospective study included 4820 pregnant women".
Materials and Methods
Point 2:
- line 87: The authors informed that multiple pregnancies were grouped as twins, triplets or multi-fetal pregnancies". The results did not include these data.
Response 2:
Previously we decided to categorize pregnancy as singleton, twins, triplets or multiple. But when we analysed our data, we found that there were only singleton and twin pregnancies. Therefore, the results are shown only for singleton and twin pregnancies. Unfortunately, we were short at correcting this sentence. Therefore, we have changed line 87 to “pregnancies were designated as singletons and twins”.
Point 3:
- The authors mentioned iron and calcium supplementation during pregnancy. What about low-dose aspirin prophylaxis? How many women received this therapy for prevention of PE, especially in the case of pregnant women with at least one high-risk condition (for example: previous pregnancy complicated by PE, multiple gestation, chronic hypertension, type 1 or 2 diabetes mellitus, chronic kidney disease, autoimmune disease)?
Response 3:
Unfortunately, in Nepal, measures like low-dose aspirin prophylaxis are not taken to prevent PE in pregnant women (even in high risk pregnancies). Therefore, we are not able to present the data on low-dose aspirin prophylaxis.
Point 4:
- The main limitation of this study is the fact that the authors selected few variables as potential risk factors, i.e. maternal age, parity, gestational age, supplementation (of iron and calcium), multiple pregnancy, and six maternal conditions (chronic hypertension, urinary tract infection, gestational diabetes mellitus, hypothyroidism, sub-fertility treatment, asthma). Is it possible to calculate additional parameters, for example maternal obesity, gestational weight gain, type 1 and 2 diabetes mellitus, maternal history of previous PE, etc.
Response 4:
We agree with your suggestion with regards to including parameters like maternal obesity, gestational weight gain, type 1 and 2 diabetes mellitus, and maternal history of previous PE; however, due to lack of these information in the data we used, we are not able to include these parameters in our model. Instead, we have added the lack of these parameters as one of the limitations of our study.
Results and Discussion:
Point 5:
- Table 1 - Is it true that p-value means 0.000 for the age 20-24 (if there are: 44.7% vs. 44%). Similarly, Table 2 - p-value 0.002 for the age 20-24 (?)
Response 5:
The p-value that we showed for the age range 20-24 is not true. Our sincere apologies for that typing error.
Point 6:
- line 130 - Age 15-19 years is a protective factor in the study, so this variable could not be indicated as a risk factor of PE.
Response 6:
We agree with your suggestion and removed that from risk factor of PE.
Point 7:
- line 130 as well as lines 183-185 - lower gestational age in the PE group could not be indicated and discussed as a risk factor of PE. This is a consequence of PE. If the diagnosis is made, the definitive treatment is delivery to prevent development of maternal or fetal complications from disease progression. Delivery results in eventual resolution of the disease. Timing of delivery is based upon a combination of factors, including disease severity, maternal and fetal condition, and gestational age. Could you specify the time of PE diagnosis? Could you include the percentage of severe PE in the PE group?
Response 7:
We agree with your suggestions and made some changes. Also removed lower gestational age from risk factor.
There was no accurate differentiation between PE and severe PE in the patient’s chart. We apologize that we could not able to include the percentage of severe PE.
Reviewer 2 Report
This is a well written and interesting paper, and relevant to the topic area.
I only have one suggestion, that the definition of pre-eclampsia (starting with line 33) is expanded, as it arguably doesn't always have to include proteinuria (for example, see this recent BMJ best practise paper (https://bestpractice.bmj.com/topics/en-gb/326).
There is also a typo within line 162 where the "the" before Nepal should be removed.
Otherwise this is a well researched paper, with an interesting and thorough discussion section.
Author Response
Dear reviewer,
Thank you for your thoughtful feedback. We have responded to your all the comments below and have made significant changes to the manuscript.
Response to Reviewer 2 comments
Point 1:
I only have one suggestion, that the definition of pre-eclampsia (starting with line 33) is expanded, as it arguably doesn't always have to include proteinuria (for example, see this recent BMJ best practise paper (https://bestpractice.bmj.com/topics/en-gb/326).
Response 1: We have accepted your suggestion to expand the definition of preeclampsia based on provided reference.
Point 2: There is also a typo within line 162 where the "the" before Nepal should be removed.
Response 2: We have accepted your suggestion to remove “the” before Nepal.
Reviewer 3 Report
Thank you for allowing me to review this article. The subject is very important and of general interest. Preeclampsia is a problem with a high incidence and that can have serious consequences for the health of the mother and the baby. The authors should answer some questions that are not clear.
-In the Abstract you must specify that these are adjusted OR. (aOR), no crude OR (OR). It can lead to confusion in the reader
-Could you clarify why maternal age was stratified in those ranges? (line 83) and Could you clarify why this classification of gestational age? (line 85). In both cases they are continuous variables.
-By not being digitized medical records, is there any possibility that some patients' medical records have not been included for some reason (lost, unavailable, etc.)?Could this have influenced the results? How could that influence have been?Should clarify more data collection.
-In the heading of table 1 there is an error, it is not univariable analysis, two variables are associated so we would be faced with a bivariable analysis
-Why is the maternal age range 20-24 years chosen as a reference?
-Could you explain what they mean by supplementation? Iron and calcium supplements? and iodine? And folic acid? (table 1) Why are these supplements chosen? Is there a certain universal supplement in Nepal for all pregnant women?
-In the results, although there are changes in the multivariable analysis, it should be noted and emphasize that there is no significance
-The authors emphasize that the quality of the data can be improved. There are data that are missing and that would be interesting such as the maternal smoking habit, history of preeclampsia in previous pregnancies, etc. that can influence the incidence of preeclampsia. Could this have influenced the results? How could that influence have been?
-The references are old, most of them are over 5 years old. They should be updated
Author Response
Dear reviewer,
Thank you for your thoughtful feedback. We have responded to your all the comments below and have made significant changes to the manuscript.
Response to Reviewer 3 comments
Point 1:
-In the Abstract you must specify that these are adjusted OR. (aOR), no crude OR (OR). It can lead to confusion in the reader
Response 1:
Yes, we agree with your suggestion and hence specified that these are adjusted OR.
Point 2:
-Could you clarify why maternal age was stratified in those ranges? (line 83) and Could you clarify why this classification of gestational age? (line 85). In both cases they are continuous variables.
Response 2:
Through different literature search we found that young maternal age (15-19 years) (ref. no 8, 11) and advanced maternal age (30-34, and ≥35 years) (ref. no. 17, 18) were associated with the increased risk of maternal and fetal complications. Therefore, we stratified maternal age in these age range.
Similarly, early gestational age (<37 weeks) and late gestational (>41 weeks) increased the risk of mother and fetal adverse outcome (ref. no. 19, 20). Gestational age from 37-41weeks is considered as the normal period. Therefore, we chose this classification for gestational age.
Also, Classification is based on other conducted studies in developed and developing countries.
Point 3:
-By not being digitized medical records, is there any possibility that some patients' medical records have not been included for some reason (lost, unavailable, etc.)? Could this have influenced the results? How could that influence have been?Should clarify more data collection.
Response 3:
Yes, we agree with your comment. Although the data is non-digitalized, we were very careful in making sure that all the available data was entered correctly. The data was first entered by the first person and then assessed for correctness by the second. We did a descriptive analysis to make sure that each category has less than 4% of missing data. Despite our sincere efforts to lessen missing data, there may be some errors because of the non-digitalized nature of the data, and we have pointed this out as one of the limitations of this study.
Point 4:
-In the heading of table 1 there is an error, it is not univariable analysis, two variables are associated so we would be faced with a bivariable analysis
Response 4:
Yes, we agree with your suggestion and have changed it.
Point 5:
-Why is the maternal age range 20-24 years chosen as a reference?
Response 5:
The age range 20-24 years have a highest number of participants compared to other age group.
Additionally, during literature search we found studies that have selected maternal age 20-24 years as reference group. Therefore, on the basis of these studies we have also chosen age 20-24 as reference group.
Point 6:
-Could you explain what they mean by supplementation? Iron and calcium supplements? and iodine? And folic acid? (table 1) Why are these supplements chosen? Is there a certain universal supplement in Nepal for all pregnant women?
Response 6:
In this study, supplementation means iron and calcium supplements, we have mentioned it in data collection section. But we will clarify it in tables too. In Nepal, there is protocol that all the pregnant women should take iron and calcium supplements after first trimester.
Point 7:
-In the results, although there are changes in the multivariable analysis, it should be noted and emphasize that there is no significance
Response 7:
We agree with your suggestion and added few sentences to this.
Point 8:
-The authors emphasize that the quality of the data can be improved. There are data that are missing and that would be interesting such as the maternal smoking habit, history of preeclampsia in previous pregnancies, etc. that can influence the incidence of preeclampsia. Could this have influenced the results? How could that influence have been?
Response 8:
We agree with your suggestion with regards to including parameters like maternal smoking habit, maternal history of previous PE; however, the data we used lacks the information on these parameters, therefore we are not able to include these parameters in our model. Instead, we have added the lack of these parameters as one of the limitations of our study. We believe that the risk factors that we identified in this study would not have changed because the risk factors that we identified like primiparity, gestational age may not be strongly correlated to maternal smoking habit. Therefore, the identified risk factors would still be independently associated with the outcome.
Point 9:
-The references are old, most of them are over 5 years old. They should be updated
Response 9:
Most of the references that are older than 2014 are the ones that present the studies related to preeclampsia conducted in developing countries. Little studies have been conducted about various aspects of preeclampsia in developing countries. Therefore, unfortunately we are not able to update the references with regards to the studies of preeclampsia conducted in developing countries. However, we have added one reference.
Tranquilli, AL.; Dekker, G.; Magee, L; Roberts, J.; Sibai, BM.; Steyn, W.; Zeeman, GG.; Brown, MA. The classification, diagnosis and management of the hypertensive disorders of pregnancy: a revised statement from the ISSHP. Pregnancy hypertension. 2014, 4(2).
Round 2
Reviewer 1 Report
Dear Authors,
Thank you for attempting to address my concerns.
I accept the manuscript in the present form.
Author Response
Dear Reviewer,
Thank you for accepting our manuscript. We will format the manuscript in this present form.
Thank you again!!
kind regards,
Seema Das